# Combined CNN and RNN Neural Networks for GPR Detection of Railway Subgrade Diseases

**DOI:** 10.3390/s23125383

**Published:** 2023-06-06

**Authors:** Huan Liu, Shilei Wang, Guoqing Jing, Ziye Yu, Jin Yang, Yong Zhang, Yunlong Guo

**Affiliations:** 1School of Geophysics and Information Technology, China University of Geosciences, Beijing 100083, China; 3010180021@cugb.edu.cn (H.L.);; 2Railway Engineering Research Institute, China Academy of Railway Sciences Co., Ltd., Beijing 100081, China; 3Infrastructure Inspection Research Institute, China Academy of Railway Sciences Co., Ltd., Beijing 100081, China; 4School of Civil Engineering, Beijing Jiaotong University, Beijing 100044, China; 5Institute of Geophysics, China Earthquake Administration, Beijing 100081, China; 6Faculty of Civil Engineering and Geosciences, Delft University of Technology, 2628 CN Delft, The Netherlands

**Keywords:** ground-penetrating radar, GPR, CNN, RNN, subgrade anomalies

## Abstract

Vehicle-mounted ground-penetrating radar (GPR) has been used to non-destructively inspect and evaluate railway subgrade conditions. However, existing GPR data processing and interpretation methods mostly rely on time-consuming manual interpretation, and limited studies have applied machine learning methods. GPR data are complex, high-dimensional, and redundant, in particular with non-negligible noises, for which traditional machine learning methods are not effective when applied to GPR data processing and interpretation. To solve this problem, deep learning is more suitable to process large amounts of training data, as well as to perform better data interpretation. In this study, we proposed a novel deep learning method to process GPR data, the CRNN network, which combines convolutional neural networks (CNN) and recurrent neural networks (RNN). The CNN processes raw GPR waveform data from signal channels, and the RNN processes features from multiple channels. The results show that the CRNN network achieves a higher precision at 83.4%, with a recall of 77.3%. Compared to the traditional machine learning method, the CRNN is 5.2 times faster and has a smaller size of 2.6 MB (traditional machine learning method: 104.0 MB). Our research output has demonstrated that the developed deep learning method improves the efficiency and accuracy of railway subgrade condition evaluation.

## 1. Introduction

Railway subgrade is a critical component of the railway system, essentially providing stable support for tracking (ballasted track, slab track), thus ensuring safe train operation [1]. However, with increased intensive high speed train loading, axle load, and traffic volume, the subgrade has undergone unacceptable rapid degradation, leading to frequent maintenance activities [2]. Subgrade defects (also as diseases and anomalies) are hazardous because they lead to rapid track degradation (such as track geometry irregularities, ballast differential settlement, sleeper damages, etc.), affecting ride comfort and safe train operations. In some situations, subgrade issues may even pose a risk to the train derailment [3].

Ground-penetrating radar (GPR) is a non-destructive technique that has been extensively used in various geophysical applications, including the inspection of transportation infrastructure, the evaluation of buried utilities, and the study of archaeology [4,5,6,7,8,9]. GPR can provide comprehensive information on the condition of the subgrade without excavation, making it a valuable tool for infrastructure managers to monitor and maintain the safety of train operations.

GPR was first applied in railway infrastructure diagnostic studies in the 1990s, using ground-coupled antennas, and, later, high-frequency air-coupled horn antennas were used with the advantage of non-contact testing at faster data acquisition speeds [10,11]. Until now, the hardware of GPR applied for railway infrastructure is mature. However, the software for processing and interpreting railway infrastructure GPR data still need improvements. One main reason is that the size of GPR data is growing exponentially, as the inspection speed, data acquisition rate, and railway line mileage have all dramatically increased. In particular, the process of applying GPR to the railway subgrade is still at the early stage of data processing. Further developments on the automated GPR data process and interpretation must be developed for rapid railway subgrade defect inspection (defect identification), finally achieving the goal of real-time railway infrastructure health monitoring.

The current GPR-based subgrade defect identification methods rely on manually designed features; afterwards, machine learning methods were used for classification. Examples are given as follows.In [12], the amplitude spectral characteristics at frequency inflection points were used to classify three types of railway ballast with a support vector machine (SVM) approach, achieving a classification accuracy of 99.5%.In [13], two-dimensional signal features were extracted, including energy and variance, as well as histogram statistical features such as mean, standard variance, smoothness, third-order moments, consistency, and entropy for radar images and structure image samples of typical subgrade defects. They then constructed a classification recognition model based on SVMs, achieving a recognition accuracy of over 85%. In [14], Hou divided radargrams of railway subgrade defects into blocks, extracted their demixing points, energy, and variance, and established optimal sparse radar features based on the L1 minimum norm method. Fuzzy C-means (FCM) and generalized regression neural network (GRNN) algorithms were used to identify railway subgrade defects, and the results showed that the classification accuracies of the sinkhole, mud-caking, and settlement were 100%, 100%, and 59.1%, respectively. 


Deep learning is an important improvement in machine learning for GPR data. With the rapid development of computer technology, deep learning algorithms, which rely on powerful computing power, have been vigorously developed for NDT infrastructure health monitoring [15,16,17,18]. CNN and RNN have the ability to learn the data structure information, and the dependencies contained between data elements have obvious superiority in target recognition and classification [19,20].
Besaw et al. [21] used CNN to classify potentially hazardous explosives below ground from GPR data, eliminating the feature selection step and demonstrating that high-precision subsurface anomaly identification can be achieved by combining neural network and ground-penetrating radar techniques.In [22], a unique Cascade R-CNN target detection framework was developed by employing 1030 annotated overturning GPR images to identify mud-pumping defects, achieving an average accuracy of 43.7%. Similarly, in [23], an LS-YOLOv3 network structure model was developed using 403 images of subgrade defects, including 261 GPR images of mud pumping and 279 GPR images of subsidence, achieving real-time fault detection with an average accuracy of 82.67%Kang et al. [6] used several B-scan maps and horizontal slice maps to form a grid image as a dataset for the GPR image feature recognition of urban cavities and pipelines using a pre-trained AlexNet model for migration learning.


CNNs have a powerful feature extraction capability, but it is not applicable for time-series GPR data. Because of this, CNNs cannot learn time-series correlation to extract the dependencies embedded in the data, while RNN networks can compensate for this shortcoming [24].McLaughlin et al. [19] proposed an RNN in order to train feature extraction networks for human re-identification, combining CNN and RNNs structures. For a video sequence consisting of a full body image of a human, each image is passed through the CNN to generate a vector, which is a vector representation of the activation mapping of the CNN output layer. The vector is then passed to the recurrent layer as an input, where it is projected into a low-dimensional feature space and combined with information from previous moments.In order to improve the recognition accuracy, Xu et al. [25] proposed the convolutional gated recurrent neural network (CGRNN). The model first uses CNN as a feature extractor, and then the extracted robust features are fed into a bidirectional gated recurrent unit (BGRU). Since the GRU can only utilize historical information, the model uses the BGRU to learn long-term audio patterns in order to utilize future information.


The combination of CNN and RNN utilizes richer information in extracting target features, which is conducive to a better description of the target, thus improving the target detection and recognition performance. However, the combination of CNN and RNN has not been studied much in ground-penetrating radar data, especially for railway subgrade. So, it is very promising, beneficial, and particularly novel to combine CNN and RNN to process the GPR data of railway subgrade for defect recognition.

Based on the above-mentioned, this paper focuses on deep neural network (DNN) detection methods for subgrade defects using raw GPR data. We introduce a novel DNN model that employs a multi-layered one-dimensional CNN to automatically learn feature functions from signal channel waveforms, resulting in a model with fewer parameters and a faster run time compared to previous DNN models that use two-dimensional CNNs. Additionally, we incorporate a multi-layer RNN to process multiple channel features from the CNN, making the model more consistent with B-scan data. We first present the design of our CNN-RNN (CRNN) model and then demonstrate its performance using manually labeled B-scan data. Our results show that the CRNN achieves comparable accuracy to Faster RCNN at a higher speed and with a smaller model size.

## 2. Methodology

### 2.1. GPR Principles

The principles of GPR are based on the propagation and reflection of high-frequency electromagnetic waves, typically in the range of 10–3000 MHz, through the subsurface. When a GPR antenna transmits a pulse of electromagnetic wave energy into the ground, some of the wave energy is reflected on the surface by subsurface interfaces, such as changes in soil or rock types, voids, or buried objects. The reflected wave energy is then detected by the same antenna or a different one, and the signals are recorded and processed to create an image of the subsurface [26].

Specifically, the waveform of GPR signals varies depending on the dielectric properties and geometry of the subsurface medium. By analyzing properties such as two-way travel time and amplitude changes in the received signal, geophysicists can deduce the depth, shape, and other characteristics of the subsurface target body, as shown in Figure 1.

The amount of electromagnetic wave energy reflected by the internal structural layer of the railway substructure (ballast layer, sub-ballast layer, and subgrade) depends on the differences in dielectric constants between the media of each layer [27,28]. As the GPR antenna moves along the track, it records the motion properties of the reflected signal, including dual-range travel duration, amplitude, and phase [29].

The GPR system works by synchronously transmitting and receiving electromagnetic waves. As the antenna moves along the track, it creates a series of scan lines (A-Scans). The A-Scans from each measurement point are gathered together based on the acquisition interval, resulting in a GPR image of the railway substructure, especially the subgrade. This image visually represents changes in the physical properties of the railway structural layer interface and the subsurface medium, which can be used to identify and extract information necessary to understand the possible defects in the railway substructure [30].

### 2.2. GPR Data Acquisition and Processing

The RIS GPR, developed by the Italian company Ingegneria dei Sistemi (IDS), is utilized to acquire GPR data, as demonstrated in Figure 2. The radar comprises a three-channel antenna group (tri-band antenna system), central control system, host radar, signal display instrument, range finder, and transmission cable. The antenna group is installed under the inspection train (developed by the China Academy of Railway Science), 30 cm above the ballast surface. The acquisition time window is set to 60 ns, with 512 sampling points and a trace interval of 11.25 cm. 

Upon system startup, the Doppler rangefinder commands the radar system to transmit pulse signals at predetermined intervals. The collector then records the radar wave signals reflected from the structural layer of the roadbed. The received signal is displayed on the screen in real time, revealing the location of each railway substructure layer.

In GPR measurements, a wideband signal is often used to capture numerous interfering signals simultaneously, resulting in more effective reflected wave characteristics. However, to extract subgrade defect information using AI models, it is essential to eliminate these interfering signals to enhance the data’s signal-to-noise ratio (SNR) [31]. Interference signals can come from three aspects [32]: First, the interference signal from the device system itself includes antennas, cables, and connectors inside. This interference is inevitable. The second is an interference from other signal sources, such as radio and television transmissions, communication signals, etc. The last one is from ground or underground/subterranean interference. This type of interference is caused by transmitting and receiving antennas that directly couple waves, or form ground-reflected waves. Due to the inhomogeneous underground medium and the occurrence of strong scattering, bypass waves are superimposed on the formation of interference waves.

Therefore, it is very necessary to pre-process the GPR data. In this study, the data were pre-processed, using the GR processing software developed by the China University of Mining and Technology (Beijing), in six steps: Sampling. Before filtering the original data, discrete sampling is required, and the sampling process needs to satisfy the sampling theorem; otherwise, the data spectrum will be mixed and thus generate false frequencies. The sampling theorem in the frequency domain equation is $\omega_{s}=2\omega_{N}\geq 2\omega_{max}$, where $\omega_{s}$ is the sampling frequency, $\omega_{N}$ is Nyquist frequency, and $\omega_{max}$ is the highest frequency of the signal [33]. Zero-line correction. The zero-line setting is mainly carried out by cutting the air layer to a fixed threshold value, which is set at the most stable point on the electromagnetic wave trajectory A-scan. Depending on the type of antenna and the center frequency, setting the appropriate threshold position along the A-scan reflects the accuracy of the results, and this threshold can be summarized as (1) the initial arrival of the wave; (2) the position of the first trough; (3) the position of the first wave crest location of the zero amplitude value between the first wave crest and the trough; (4) the location of half the amplitude value between the first wave crest and the trough; (5) the position of the first wave crest [34].Gain setting. Using automatic gain control (AGC). When the signal is strong, its gain automatically decreases and when the signal is weak its gain automatically increases. It can ensure the uniformity of strong and weak signals and facilitate the tracking of effective waves [35].FIR bandpass filtering. The bandpass filter works by cutting off the fringe bands from the spectrum of GPR data. The filter consists of two filters, a high-pass filter and a low-pass filter, which modify the ground-penetrating radar signal by removing the low- and high-frequency components of the spectrum. As a rule of thumb, a bandwidth of 1.5 times the survey center frequency can be used initially [32,35].Running average filters. The background noise of the GPR signal is high, and the autoregressive sliding average spectral estimation (ARMA) in modern spectral estimation is used to analyze and identify the non-smooth signal of the effective reflected signal, which can extract the signal features at a low signal-to-noise ratio with a high accuracy of spectral estimation [36].


After pre-processing, the inevitable and random interference signals in the GPR image are suppressed, increasing the SNR. The variations of dual trip time (or depth) and mileage are reflected in the GPR image features. These image features are the reflected wave amplitude, waveform, phase, and spectrum of the inter-layer interface of the subgrade, as well as the subgrade defect. A collection of typical GPR images for various defect types is shown in Table 1.

### 2.3. Training and Testing Dataset

Our approach is designed to process original pre-processed GPR data. The original signal files record a series of binary data called A-scan. We used MATLAB to convert the processed data into decimal data based on the arrangement of GPR raw data, which was then saved in MAT format. The MAT format dataset is a two-dimensional matrix, where n is the number of channels (or the number of A-scan) and m is the perpendicular length of waveform data. The echo data F can be represented by an m×n matrix as follows:$$F=\left[\begin{array}{cccc} x_{11} & x_{12} & \cdots & x_{1n}\\ x_{21} & x_{22} & \cdots & x_{2n}\\ \vdots & \vdots & \ddots & \vdots \\ x_{m1} & x_{m2} & \cdots & x_{mn} \end{array}\right]$$
where $x_{\mathrm{ij}}$ represents the response recorded at the jth channel and the ith sampling point.

We selected 3000 channels of GPR pre-processed data and inverted them into an image, as shown in Figure 3 (left figure). The number of subgrade defects among these 3000 GPR data was counted, as shown in the right graph in Figure 3. 

Additionally, to test the efficiency of the CRNN network model, we compared it with object detection network models such as Faster R-CNN and Yolov3 models. The two-dimensional matrix dataset was converted into GPR images using an IDS signal treatment software named SRS DPA basic 02.02.004, as shown in Figure 4.

The object detection formula is ximg∈ℝH×W×3. H is the height of images for which the sampling point N exists in GPR data. W indicates the width of an image by the number of channels of GPR data. To compare CRNN and object detection methods, the inputs of an image are ximg and xGPR.
xi,jGPR=13∑kxi,j,kimg

We note that $x^{\mathrm{GPR}}$ is different to $x^{\mathrm{orig}}$. $x^{\mathrm{GPR}}$ is the grayscale map of an image $x^{\mathrm{img}}$, which includes a boundary output by SRS DPA software. The bit-rates of $x^{\mathrm{GPR}}$ are 8 and 24 of $x^{\mathrm{orig}}$. 

### 2.4. CRNN Network Structure

The CRNN is a fusion network that combines CNN and RNN to process different GPR features. The network architecture is shown in Figure 5. CNNs were first proposed by Le Cun et al. as a highly nonlinear mapping method in supervised learning mode for establishing the connection between target samples and inputs [37]. RNNs are neural networks with recurrent connections that can model sequence data for sequence recognition and prediction [38]. They can also store information using recurrent iterative functions, capture contextual information well, and achieve transient dependency learning. The CRNN network takes advantage of both CNN and RNN to process GPR data, making it an effective tool for detecting defects in the subgrade.

The CRNN network employs one-dimensional convolution to process the original waveform directly. To enhance the convergence speed of the network, we incorporate a batch normalization layer into the convolutional network. The calculation of a single-layer network can be expressed as the following formula:$$\begin{array}{c} x^{k+1}=x^{k}*w^{k}+b^{k}\\ {\hat{x}}^{k+1}=\gamma \frac{x^{k+1}-\mu^{\;}}{\sqrt{\sigma^{2}+\varepsilon }}+\beta^{k}\\ y^{k+1}=\mathrm{RELU}\left({\hat{x}}^{k}\right) \end{array}$$
where ∗ represents the convolution operation, and w^k^, b^k^ are the trainable parameters in the convolution layer. μ and σ represent the mean and standard deviation of the data, while γ and β are the trainable parameters in the normalization layer. The addition of normalization in the convolution layer can effectively accelerate the model’s convergence. Additionally, the trainable parameters added to the normalization layer can be used to calculate the data’s magnitude and mean value.

For downsampling, we add a stride of 2 to the convolutional layer instead of using a pooling layer in the model. The original single-channel waveform contains 512 sampling points. After 9 convolutional layers with a stride of 2, we end up with a feature vector of length 9F, where F is the number of filters in the convolutional layers. This feature vector is extracted from the waveform and is used for further processing.

Waveform features are typically manually created using techniques such as wavelet transform and frequency domain transform, while CNN features are automatically extracted from training data [20]. This suggests that it is possible to obtain characteristics automatically, without physical effort.

RNNs can be viewed as a collection of interconnected networks, where the output of one network is fed into the next in a chained design [39]. As a result, both the input and output of the network are determined before the network has an impact on the output of the subsequent network. In contrast, the input and output of CNNs are independent of each other. RNNs have a “memory” that retains all the computed information. In our model, RNN layers are used to process features from a single waveform, where the feature from the kth station is hk∈ℝ10F. There are N channels in total, and so the input to the RNN is h∈ℝN×10F. Our model’s RNN is a unidirectional RNN, meaning it can process both the current and prior channel features and produce accurate detection results, represented as ok. Unidirectional RNN predicts the outcome as
p(ok|xk,xk−1,⋯,x1)

This implies that the detection result depends on the previous waveform. Processing the B-Scan in time is enabled by the unidirectional RNN structure, which is simple, easy to train, and achieves high accuracy.

The final step in our CRNN model involves adding two fully connected output layers on top of the RNN output. The first output layer generates the classification result ck∈ℝ2, which determines the type of the kth waveform. We have two types of waveforms: normal and anomalous. The softmax function is applied to constrain the output ck:cki=expyk,i∑mexpyk,m

Here, $y_{k,i}$ is the original output of the classification layer, which is derived from the kth waveform of the ith type.

The second output layer generates the regression result rk∈ℝ2, which indicates the upper and lower positions of an anomalous waveform. The sigmoid function [40] is applied to constrain the output rk:rki=512⋅sigmoidyk,i

We use the sigmoid function to limit the boundaries of the regression output so that the upper and lower positions do not exceed the length of the waveforms. This enhances the robustness of the regression output [41].

### 2.5. Anomaly Detection

The outputs of CRNN are regression and classification. To obtain a box position of anomaly, some post-processing is required. The two types of output are shown in Figure 6. The vertical red line represents the left border are represented by $b_{x1}$ and $b_{x2}$, respectively, while $b_{y1}$ and $b_{y2}$ indicate the upper and lower boundaries. The predictions of the CRNN, represented by r_k_ and c_k_, are used to determine the upper and lower boundaries and the type of anomaly in the kth channel. The start and end of an anomaly type in ck can be used to constrain the left and right boundaries of the anomaly, i.e., $b_{x1}$ is the start of an anomaly type and $b_{x2}$ is the end of it. The upper and lower boundaries are determined by the mean of the anomaly type, $x_{i}=E\left(r\right),i=1,2$.

### 2.6. Training of CRNN

The CRNN loss function includes both classification and regression errors, and it is expressed by the following formula:$$L=-\frac{1}{N}\underset{k}{\sum }\log c_{k,d}+\frac{1}{N}\underset{k}{\sum }w_{k}\underset{i}{\sum }{\left(r_{k,i}+{\hat{r}}_{k,i}\right)}^{2}$$
where d represents the type of the current channel, and ${\hat{r}}_{k,i}$ is the manually labeled boundary. The weight of the regression loss is represented by $w_{k}$, which is set to 0.0 when there is no anomaly, and it is set to 1.0 when the kth channel is anomalous.

To optimize the CRNN structure, the Adam algorithm [42] was used instead of stochastic gradient descent. The initial learning rate was set to $1\times 10^{-4}$, and after 1000 iterations, we reduced it to $1\times 10^{-5}$ to ensure a more stable convergence. Figure 7 shows the visualization of the CRNN loss function.

Our network structure here is training on Titan RTX with a 24G memory and a Threadripper 3975WX CPU. It needs 12 h to train our model. The deep learning framework is PyTorch 1.9 and CUDA 11.3. The loss function ranges from 12.5 to about 1.5 after 16,000 iterations. 

For the initial 2500 iterations of the iterative process, we used a $1\times 10^{-4}$ learning rate to warm up the network, which can prevent the instability problem during the iterative process, and the learning rate in the subsequent iterations was chosen as $1\times 10^{-3}$, and the convergence speed became faster. Since the data within each batch are randomly selected, there are oscillations in the iterations, but the overall trend remains constant after 14,000 iterations, with a small increase in the loss function around 13,500 iterations, due to re-training after program termination.

## 3. Case Study

### 3.1. Detection Result on Real Railway Subgrade GPR Data

Our methods can process the original GPR waveform. We selected one typical Chinese existing railway line as the test dataset, which consists of eight data packets. Every GPR data packet includes 266,666 channels. We selected 1000 channels of the fifth data packet to test the detection result, as shown in Figure 8.

The results from Figure 8 demonstrate that the left and right boundaries of an anomaly are constrained by the probability of the classification output. Anomalies start with a probability greater than 0.5 and end with a probability smaller than 0.5 (as shown in Figure 8b). The upper and lower boundaries are determined by the regression output, and the mean of the regression is used to establish the boundary (as shown in Figure 8c). The anomaly box can be determined by combining the regression and classification results (as shown in Figure 8d). Overall, the detection results are in close agreement with the manually labeled data.

During the detection process, each channel classifies anomalies, and the lateral accuracy depends on the channel spacing. In Figure 8b, we observe that the probability of anomalies is not constant, and some anomalies have a lower confidence level due to being offset in the homophase axis in the GPR. However, CRNN can synthesize data from other channels, which enables it to identify the data belonging to an anomaly by combining information from other channels. This is one of the advantages of RNN modeling compared with single-channel data analysis.

Figure 8c shows that even the normal tract has predicted upper and lower boundaries due to the lack of constraints on the training data. Therefore, some randomness may occur in the prediction of the normal tract. However, since the regression output does not constrain the horizontal position of the abnormality, it needs to be combined with the classification output in Figure 8b to determine the lateral boundary. The combination of these two approaches results in the accurate detection of the abnormal position, as shown in Figure 8d.

Our detection model is unidirectional and can be applied to GPR data of any length in the B-scan direction. To test the accuracy of our model, we utilized the full GPR section, which consists of 26,666 channels. We evaluated our model using four indexes:
Precision (P). This is the rate of successfully detected channels. It is calculated as $P=\frac{\mathrm{TP}}{\mathrm{TP}+\mathrm{FP}}$.Recall (R). This is the rate of the detected anomalous channels out of all the anomalous channels in the dataset. It is calculated as $R=\frac{\mathrm{TP}}{\mathrm{TP}+\mathrm{FN}}$.Mean error of the four boundaries. This indicates the bias of our model.The standard deviation of the four boundaries. This measures the variability in our model’s performance.The detection model is a single-way model.


TP samples are channels that are both manually labeled and detected by our model. FP samples are detected by our model but not manually labeled, while FN samples are manually labeled but not detected by our model. The results of the four evaluation indicators are presented in Table 2.

Table 2 presents the results of our evaluation using different down-sampling rates ranging from one to six. To downsample, we took one channel out of each S channel. The x-direction and y-direction are in different units: the unit of the x-direction is the number of channels, while the unit of the y-direction is the number of sampling points. The error of the x-direction (left and right boundary) is approximately 20 channels, and the error of the y-direction (upper and lower boundary) is approximately 18 sampling points.

We also calculated the P–R curve for different strides and plotted it in Figure 9. The results show that our CRNN model achieves high accuracy across all strides.

By using stride, we downsampled the GPR data in the scanning direction while maintaining consistent accuracy. This allowed our model to handle GPR data of different scales and channel spacing while ensuring accuracy.

### 3.2. The Inferring Speed in GPR Data 

Inferring speed is a critical consideration when deploying a deep neural network. To ensure our CRNN model can be run on a wide range of devices, we tested it on both CPU and GPU devices, as shown in Table 3. We also transformed the CRNN model to an ONNX format, which is a lightweight DNN framework that can run faster than other models.

Since the RNN model is not easily parallelizable, its speed is similar to CPU or GPU. The inference speed of our model is approximately 2.139 s. Being a single-way model, our model can analyze GPR data instantly. The unidirectional model ensures that the current result depends only on the information from the previous channel, making it computationally more efficient and suitable for real-time processing.

As the RNN model is time-dependent, its hypothetical effect is not apparent on the GPU, but it is computationally faster on the CPU.

### 3.3. Comparison with Object Detection Models

We compared our methods with Faster R-CNN, which takes an image as the input. To enable comparison, we transformed the color image, where the height and width of the image were transformed into the GPR data format. First, we transformed the color image into gray by taking the mean of its channels:$$x_{i,j}^{\mathrm{gray}}=\frac{1}{3}\overset{3}{\underset{k=1}{\sum }}x_{k,i,j}$$

The resulting gray image had the format x^gray∈R^(H × W). However, GPR data have a different unit, where W represents the number of channels and H indicates the number of sampling points in one channel. Since CRNN requires H = 512, we resized the image x^gray to have W^’ = W 512/H channels, resulting in GPR-like data x^gpr∈R^(512 × W^’). The accuracy was tested on the zoomed data x^gpr. Since the image has padding, we retrained our model to fit the image data rather than the GPR data.

We manually labeled 2800 images, of which 2400 were used to train the models and 400 were used to test the accuracy. Both CRNN and Faster R-CNN used the same training and test datasets. The hyperparameters of CRNN were the same as for real GPR data. Faster R-CNN is an open-source project that can be downloaded from GitHub.

#### 3.3.1. Detection Result on Image Data

The detection results from CRNN and Faster R-CNN are shown in Figure 10. 

Figure 10 shows that the object detection and CRNN models produce different results. The CRNN model tends to detect small anomalies due to its reliance on the class predictions of the tracks. Therefore, if there is noise in consecutive anomalous channels that are detected as normal, the detection process may split a large anomaly into multiple small ones. To address this issue, we removed the anomalies containing less than X number of channels in the post-processing stage. In contrast, the Faster R-CNN model produces more complete results since it is based on individual anomalous image pixels rather than individual channels. The longitudinal resolution of the object detection model is arbitrary, while our model needs to be combined with RNN for processing. Hence, the number of longitudinal sampling points is limited to 512, and if it is insufficient or exceeded, manual cropping is required.

#### 3.3.2. Statics on Image Data

As shown in Table 4, our CRNN model achieves higher accuracy than Faster R-CNN when IoU is greater than 0.25 or 0.5, indicating that our model is effective in detecting railway subgrade anomalies. However, when the IoU is greater than 0.75, the accuracy of Faster R-CNN is higher than our model. This may be due to the fact that Faster R-CNN is based on individual anomalous image pixels rather than individual channels, which allow it to detect more detailed information in the image. Nonetheless, our CRNN model still performs well in detecting subgrade anomalies, and can provide valuable information for railway maintenance and safety.

Table 5 demonstrates that the CRNN model has a much smaller size compared to Faster R-CNN, making it more memory efficient for inference. Additionally, our CRNN model exhibits faster inference speeds on both CPU and GPU, allowing for deployment on a wider range of devices while maintaining satisfactory performance in railway subgrade anomaly detection.

Based on the experimental results and literature survey, this paper analyzes the advantages and disadvantages of four network models, respectively, CNN, RNN, CRNN, and Faster RCNN, as shown in Table 6.

## 4. Conclusions and Perspective

We developed a novel GPR anomaly detection model called the CRNN network, which is based on a hybrid CNN and RNN architecture. While most existing GPR detection models rely on object detection models, which are more widely used and easier to train, these image-based models do not consider the unique physical characteristics of GPR data, such as the different physical meanings of the sampling points in the B-Scan direction of the channels and waveforms [34]. This can lead to accuracy degradation since GPR data need to be converted to image data during processing.

In contrast, our CRNN model is specifically designed to account for both longitudinal and transverse physical quantities, making it more targeted than object detection models built directly using CNNs. Our one-dimensional CNN processes waveform sampling point data, while the RNN processes transverse channel data. This allows our model to directly process GPR data without the need for image conversion, resulting in a smaller and more computationally efficient model.

Our CRNN network has a size of only 2.6 MB, compared to 104 MB for the Faster R-CNN model, making it more computationally efficient. Additionally, our model is four times faster than object detection models, as it only deals with single-channel waveforms and does not need to process inter-channel data. Overall, our CRNN network provides a more accurate and efficient approach to GPR anomaly detection.

Our CRNN network demonstrates comparable accuracy to Faster R-CNN, with some differences in performance at different IoU thresholds. Specifically, our model performs better at IoU ≤ 50% due to its ability to detect anomalies in channels during the detection process. However, it may detect some large anomalies as multiple small anomalies due to noise, resulting in more small anomalies at the anomaly boundary that require filtering based on expert experience. Additionally, the starting position of anomalies may not always be clear, leading to alternating normal and anomaly readings at the boundary.

During deployment in a real production environment, our lightweight model can save computational resources as it does not require converting GPR data into images, or post-processing for anomaly localization. With sufficient training data, the CNN model can even process GPR data directly without filtering, further shortening the processing time. Additionally, the use of a one-way RNN model allows for real-time data processing without interception.

Our detection model is limited to determining the presence of anomalies and does not classify specific types of anomalies. More data should be added in subsequent work to improve classification accuracy. Additionally, the testing accuracy of our model may be slightly lower than other articles due to the complexity of our actual working condition data, highlighting the need for further data accumulation to improve accuracy.

## Figures and Tables

**Figure 1 sensors-23-05383-f001:**
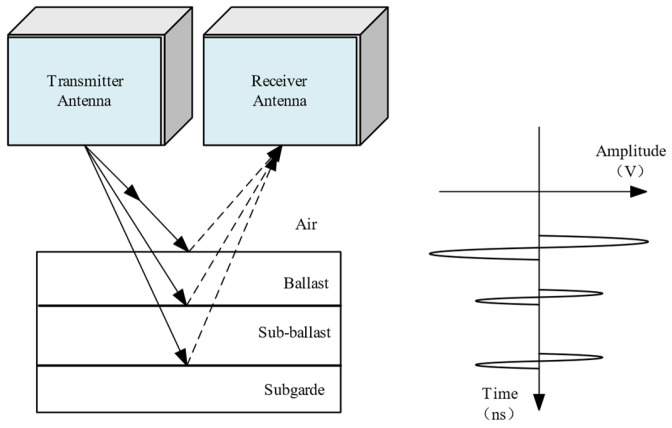
Schematic diagram of GPR detection for railway subgrade.

**Figure 2 sensors-23-05383-f002:**
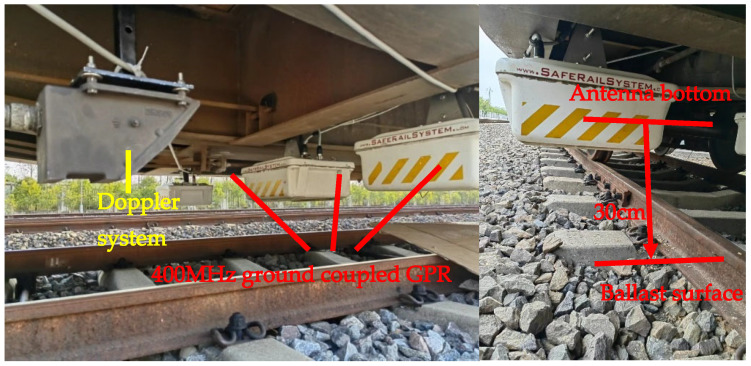
The subgrade status inspection vehicle with a three-channel GPR antenna group.

**Figure 3 sensors-23-05383-f003:**
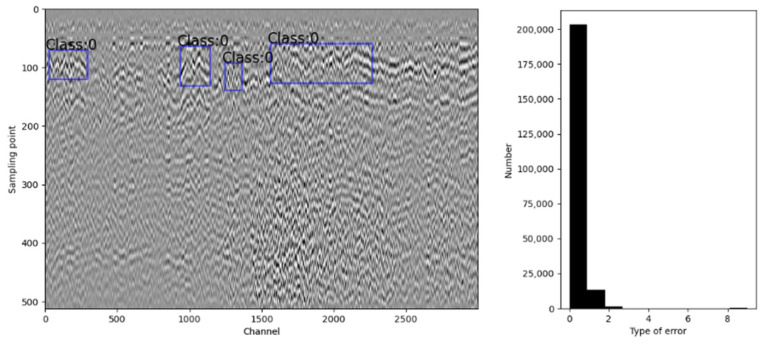
Original GPR data and its statics.

**Figure 4 sensors-23-05383-f004:**
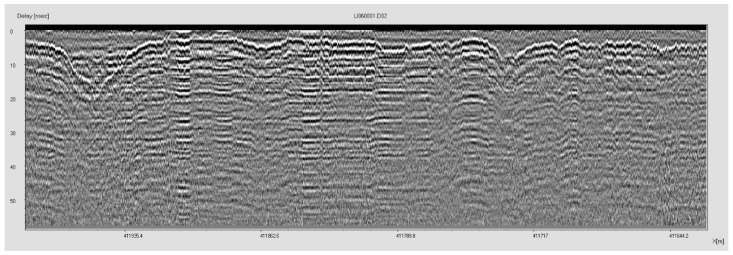
GPR image data exported by SPS DPA software.

**Figure 5 sensors-23-05383-f005:**
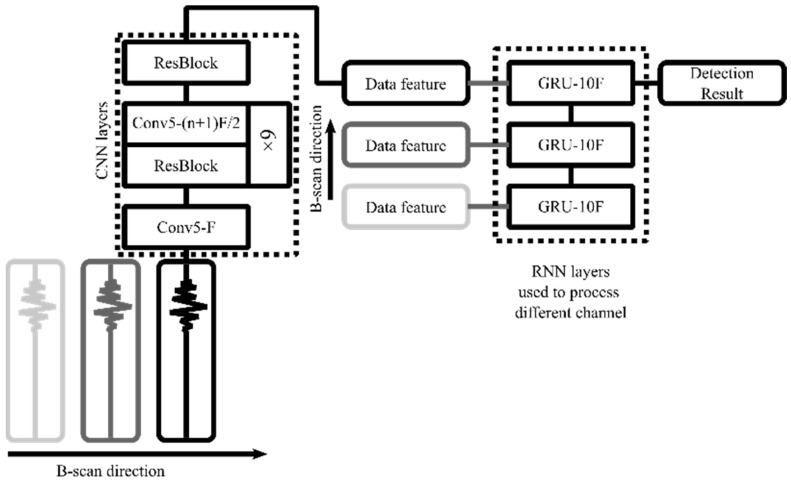
Network structure diagram of CRNN.

**Figure 6 sensors-23-05383-f006:**
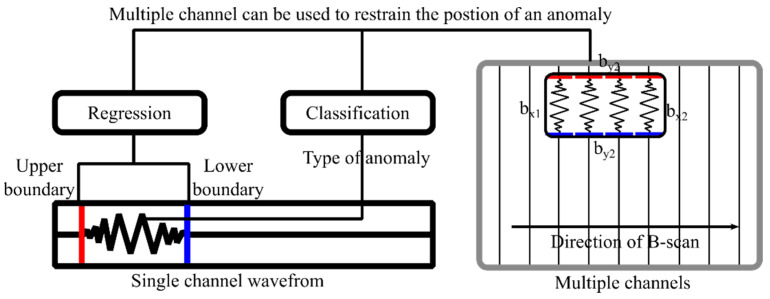
Illustration of two outputs of CRNN. The vertical red line represents the left boundary. The vertical blue line represents the right boundary. The horizontal red line represents the upper boundary. The horizontal blue line represents the lower boundary.

**Figure 7 sensors-23-05383-f007:**
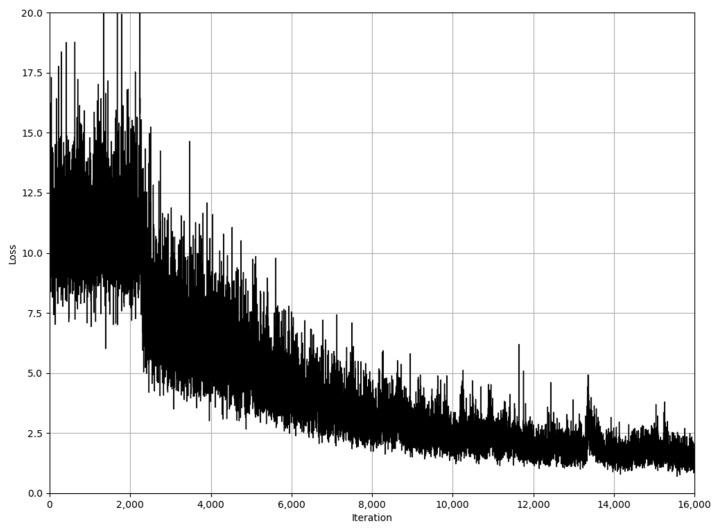
Loss function of CRNN structure.

**Figure 8 sensors-23-05383-f008:**
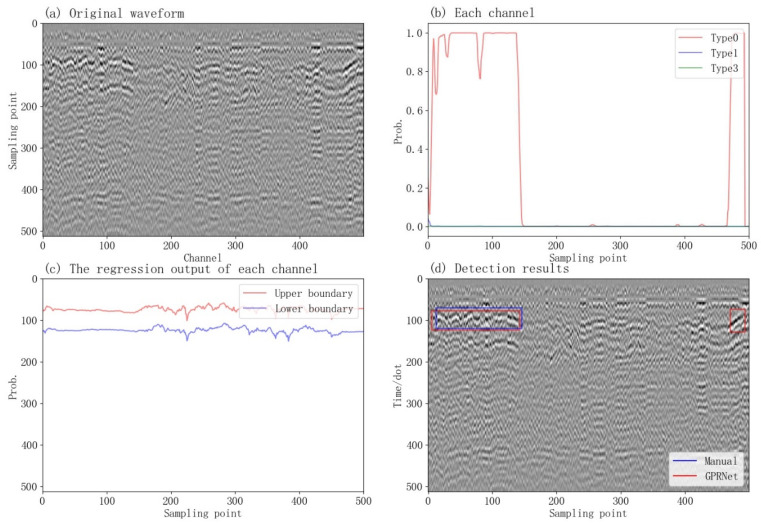
The detection result of 1000 channels from the fifth data packet. (**a**) Original waveform data. (**b**) Each channel. (**c**) The regression output of each channel. (**d**) Detection results.

**Figure 9 sensors-23-05383-f009:**
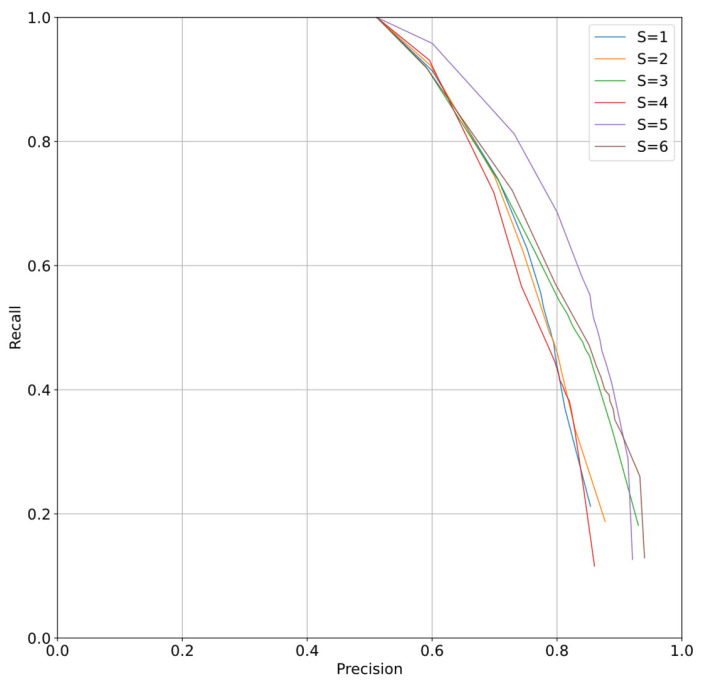
P-R curve of different strides of CRNN.

**Figure 10 sensors-23-05383-f010:**
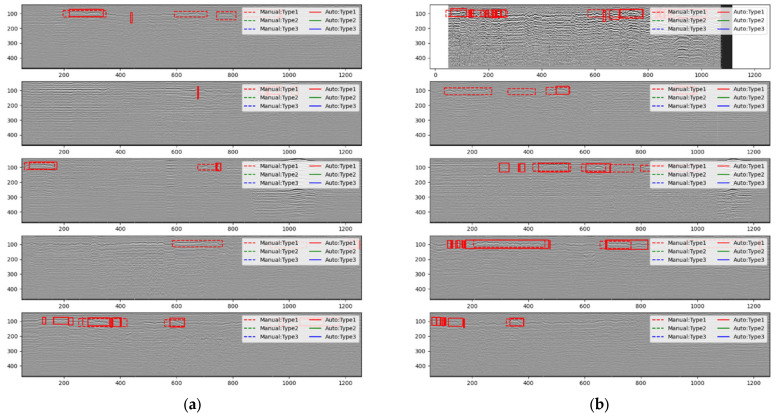
Detection results of the two models. (**a**) CRNN detection results, (**b**) Faster R-CNN detection results.

**Table 1 sensors-23-05383-t001:** Different types of railway subgrade defects.

Railway Subgrade Defect Types	Typical Defects Image	Image Features
Normal subgrade without defect	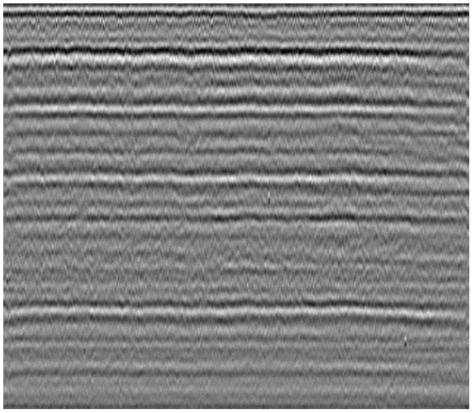	The lamellar structure is obvious. The in-phase axis is straight and continuous. The reflected energy is uniform.
Mud pumping	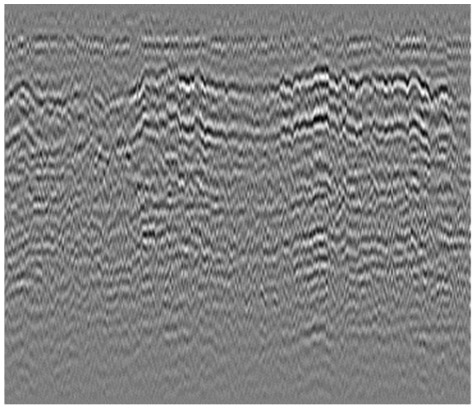	The wave group is a disorganized, discontinuous, low-frequency strong reflection shape that resembles a mountain tip or straw hat.
Settlement	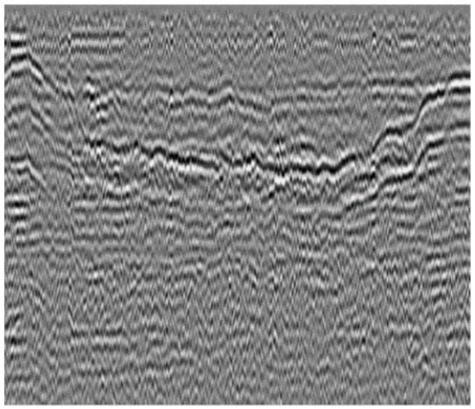	The reflection of the in-phase axis of the settlement radar image is significantly bent, with depth-downward offset.
Water anomaly	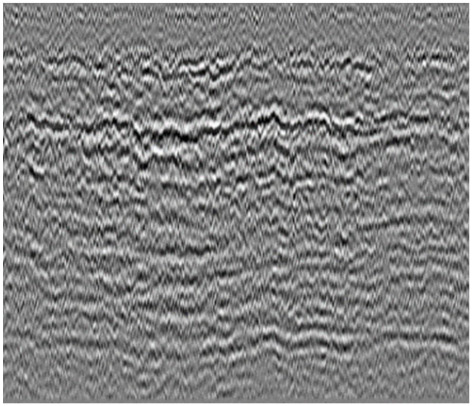	The signal is attenuated. The reflected energy of the top surface is strong. Multiple waves exit.

**Table 2 sensors-23-05383-t002:** Statistical results with different strides.

Model	P	R				
S = 1	0.79	0.74	−14.73	−40.47	30.25	48.59
S = 2	0.79	0.7	−17.67	−44.56	31.05	50.98
S = 3	0.79	0.71	−18.13	−42.86	30.4	49.75
S = 4	0.81	0.67	−20.74	−47.71	32.06	52.6
S = 5	0.76	0.73	−16.38	−39.54	28.87	48.58
S = 6	0.76	0.65	−22.41	−50.83	31.3	52.39

**Table 3 sensors-23-05383-t003:** Inferring speed (test on 266,666 channels).

Running Device	Inferring Time (ms)
CPU	1325
GPU	2139

**Table 4 sensors-23-05383-t004:** Precision and recall comparison with different methods.

Model Name	Faster RCNN			CRNN (Ours)		
	Precision	Recall	F1Score	Precision	Recall	F1Score
IoU > 0.25	0.777	0.936	0.849	0.795	0.941	0.862
IoU > 0.50	0.752	0.671	0.709	0.834	0.773	0.803
IoU > 0.75	0.527	0.260	0.348	0.422	0.158	0.230

**Table 5 sensors-23-05383-t005:** The size and training time of different models.

Model Name	Faster RCNN	CRNN (Ours)
Detected 238 images on CPU (ms)	18,000	2600
Detected 238 images on GPU (ms)	2000	500
Model Size (MB)	104	2.6

**Table 6 sensors-23-05383-t006:** The tabulation of the advantages and disadvantages of different deep models.

Deep Model	Advantages	Disadvantages
CNN	One-dimensional CNN can extract features in-depth, i.e., longitudinally, in the time dimension, generating a vector for each GPR waveform which can reduce the storage requirements [19].	Since CNN has no memory function for time series, it cannot take the input data sequence into account like RNN [24].
RNN	RNN can be regarded as a series of interconnected networks that have a chained architecture, where the output of the next network depends on both its input and the output of its predecessor network [43].	Because of the chain derivative rule and the existence of a nonlinear activation function, RNN is prone to the problem of gradient vanishing and explosion [44].
CRNN	Lightweight, the model size is only 2.6 MB. Faster inference speeds on both CPU and GPU are easy to deploy on hardware devices. No complex data processing steps are required (e.g., filtering and information compression), taking into account the physical mechanisms of GPR data and their temporal dynamic behavior.	Due to the insufficient sample number and sample imbalance, the CRNN model only determines the existence of anomalous regions, and the identified anomalies still need to be interpreted manually. When the IoU is greater than 0.75, the accuracy of the CRNN model is low, and more neurons may need to be added to improve accuracy.
Faster R-CNN	Faster RCNN, as a two-stage network, contains RPN and RCNN, and has higher recognition accuracy compared to the one-stage network. Many deep learning frameworks have a practical [45] Faster RCNN source code that is easy to use.	Although the detection accuracy of Faster RCNN is improved, the training speed of its network model is slow [46].

## Data Availability

Not applicable.

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
