# Peer review of "Combined CNN and RNN Neural Networks for GPR Detection of Railway Subgrade Diseases"

_sensors, 2023, doi:10.3390/s23125383_

Round 1

Reviewer 1 Report

The article discusses the limitations of traditional machine learning methods in processing Ground Penetrating Radar (GPR) data for railway subgrade condition evaluation and proposes a new method that combines Convolutional Neural Networks (CNN) and Recurrent Neural Networks (RNN). The results indicate that the proposed method achieves a higher precision and lower computational cost, compared to Faster R-CNN. Overall, this manuscript is well organized and written, and addresses a meaningful issue. Before publication, the authors are recommended to address the following minor comments:

1. Section 2: please check the heading of subsections 2.2 and 2.3 and renumber them.

2. There are errors in the numbering and citation of the figure, including but not limited to: Figure 5 on page 8 is wrongly cited; Figure 7 on page 10 is wrongly cited; Figure 9 on page 13 is cited incorrectly; Figure 7 on page 14 is numbered incorrectly. Please check and revise the figures and tables in the text.

3. In the introduction of the application of deep learning in GPR data processing, I suggest also to extend references to literature to recent international works.

4. In Section 2.2, the preprocessing of GPR data is mentioned. Can you provide relevant references, or give a detailed description?

5. Page 6, 1st paragraph: please describe the method of converting the pre-processed GPR data into decimal data.

6. Page 6, 1st paragraph: The MAT format dataset is a two-dimensional matrix, where n is the number of channels ... and m is the perpendicular length of waveform data. I guess m and n are from a missing equation, please add the explanation.

7. The first sentence of the second paragraph on page 8 is used as an attributive to the previous paragraph, so it is not necessary to use the paragraph format with the first line indented. Paragraph 2 on page 10 has a similar problem.

8. Title of Section 3: should it be “Case Study” instead of Results?

9. In Section 3.1, the GPR data set of Line #5 is selected for testing. What is the meaning of Line #5? Please give a description of the line and test conditions.

10. In the figures of GPR data, you may consider using the response time or buried depth as the vertical coordinate.

11. Some of the figures in this article are not clear enough. Please replot them, and unify the text format in the figures. Axes should be labeled with names so that the reader can understand the figure.

12. References [29], [30] are journal papers in Chinese, please consider marking '(in Chinese)' in the reference list. Besides, there is an error in the author name of the reference [30], please check and modify it.

Author Response

The article discusses the limitations of traditional machine learning methods in processing Ground Penetrating Radar (GPR) data for railway subgrade condition evaluation and proposes a new method that combines Convolutional Neural Networks (CNN) and Recurrent Neural Networks (RNN). The results indicate that the proposed method achieves a higher precision and lower computational cost, compared to Faster R-CNN. Overall, this manuscript is well organized and written, and addresses a meaningful issue. Before publication, the authors are recommended to address the following minor comments:

  1. Section 2: please check the heading of subsections 2.2 and 2.3 and renumber them.

The headings are double checked and corrected.

  1. There are errors in the numbering and citation of the figure, including but not limited to: Figure 5 on page 8 is wrongly cited; Figure 7 on page 10 is wrongly cited; Figure 9 on page 13 is cited incorrectly; Figure 7 on page 14 is numbered incorrectly. Please check and revise the figures and tables in the text.

The figure citation is corrected.

  1. In the introduction of the application of deep learning in GPR data processing, I suggest also to extend references to literature to recent international works.

More international references are added.

  1. In Section 2.2, the preprocessing of GPR data is mentioned. Can you provide relevant references, or give a detailed description?

The references are added and the detailed description of the preprocessing of GPR data are given.

  1. Page 6, 1st paragraph: please describe the method of converting the pre-processed GPR data into decimal data.

The description for that is added. Thank you for the advice, very helpful!

  1. Page 6, 1st paragraph: “The MAT format dataset is a two-dimensional matrix, where n is the number of channels ... and m is the perpendicular length of waveform data”. I guess m and n are from a missing equation, please add the explanation.

Yes, their explanations are added. They are the column and row of the matrix.

  1. The first sentence of the second paragraph on page 8 is used as an attributive to the previous paragraph, so it is not necessary to use the paragraph format with the first line indented. Paragraph 2 on page 10 has a similar problem.

The format is changed.

  1. Title of Section 3: should it be “Case Study” instead of “Results”?

The section head name is changed.

  1. In Section 3.1, the GPR data set of Line #5 is selected for testing. What is the meaning of “Line #5”? Please give a description of the line and test conditions.

Because the line name is confidential in China, which means in any publications the railway line name and location can not be shown. So, our apologies for the misleading name. The name has been omitted.

  1. In the figures of GPR data, you may consider using the response time or buried depth as the vertical coordinate.

It is a very good advice, in our future research we will keep digging on this direction. We have added this idea in the perspectives.

The vertical coordinates of some pictures in the article are the sampling points, and the conversion of electromagnetic wave velocity and depth is not considered. In the next work we should solve this problem by conducting digging tests in the field.

  1. Some of the figures in this article are not clear enough. Please replot them, and unify the text format in the figures. Axes should be labelled with names so that the reader can understand the figure.

Figures are modified to make it more clear.

  1. References [29], [30] are journal papers in Chinese, please consider marking '(in Chinese)' in the reference list. Besides, there is an error in the author name of the reference [30], please check and modify it.

The references are modified.

Reviewer 2 Report

1 In general, the theme is interesting but the research gap in the introduction is not visible and the urgency of the research is not sharp

2 The results section must be accompanied by relevant previous references

3 Methodology is accompanied by relevant references

4 The discussion on the results is explained more comprehensively

Author Response

1 In general, the theme is interesting but the research gap in the introduction is not visible and the urgency of the research is not sharp

More references are added to show the research gap and the urgency of the research.

2 The results section must be accompanied by relevant previous references

The relevant references are added.

3 Methodology is accompanied by relevant references

The relevant references are added.

4 The discussion on the results is explained more comprehensively

A tabulation of advantages and disadvantages for the innovative solution has been added.

Reviewer 3 Report

- Please restructure section 1. Optimise the contribution and place related work in a separate section. 

- What is the conclusion? Is the Faster R-CNN better or worse than your approach?

- The fact that Faster R-CNN is based on individual anomalous image pixels rather than individual channels, are you certain that this is the case of higher accuracy?  Which IoU is more significant for the purpose of this work?

- Please elaborate more on the limitations of your work and how they are reflected in the given problem.

- Why did you use Faster R-CNN?

Author Response

1 Please restructure section 1. Optimise the contribution and place related work in a separate section.

Section 1 has been restructured.

2 What is the conclusion? Is the Faster R-CNN better or worse than your approach?

Our CRNN network has a size of only 2.6 MB, compared to 104 MB for the Faster R-CNN model, making it more computationally efficient. Additionally, our model is four times faster than object detection models, as it only deals with single channel waveforms and does not need to process inter-channel data.

3 The fact that Faster R-CNN is based on individual anomalous image pixels rather than individual channels, are you certain that this is the case of higher accuracy?  Which IoU is more significant for the purpose of this work?

Faster Rcnn is an image-based target detection algorithm, CRNN is a classification algorithm based on ground-penetrating radar data matrix, either algorithm needs big data as support.

the raw GPR signals are directly used as inputs in our CRNN network, which reserve all useful and useless information. Compared with the architectures exploiting image, the ones exploiting A-scan have a slight advantage in defect class recognition and location computation. Because some necessary pre-processing (e.g., filtering and information compress) is conducted on GPR data for the utilization of image as the input. These pre-processing procedures sometimes lead to feature and information loss.

  1. Please elaborate more on the limitations of your work and how they are reflected in the given problem.

The use of deep learning methods for disease classification requires the support of big data. For the classification problem of railroad roadbed diseases, since roadbed diseases are affected by many external factors leading to the same kind of disease performance on ground-penetrating radar images varies greatly, although our CRNN model can learn features automatically, the data set is still far from enough. In addition, the amount of data varies greatly among different disease types, which results in large differences in the evaluation metrics of different types of diseases in our results. More diseases need to be labeled in the next work to improve the accuracy.

  1. Why did you use Faster R-CNN?

In order to prove the advantages of our model, it is compared with the Faster R-CNN model.

Our CRNN network has a size of only 2.6 MB, compared to 104 MB for the Faster R-CNN model, making it more computationally efficient.

Reviewer 4 Report

Dear Authors,

The publication is distinguished by the reference to own research with noticeable practical experience. The adopted and implemented structure of the publication corresponds well to the goal, which allowed to draw a wide range of conclusions.
Thanks.

Further information on various issues identified in the manuscript appears below:

- I propose to expand the scientific research literature to include issues in the integration of UAVs with GPR used in rail infrastructure surveys.

- It is proposed to make a tabulation of advantages and disadvantages for the innovative solution.

- Section 2 - the accuracy of the GPR in terms of monitoring the railway infrastructure should be presented - in the situation where the GPR is built into the train structure. Then we have different speeds. What are the GPR accuracies depending on the speed of the vehicle?

- The numbering of chapters and subsections needs to be improved throughout the publication. Sometimes identical subsection numbers are used, e.g.: 2.3.

- The entire text needs improvement in punctuation marks and style.

- The description of Figure 7 should not be in bold.

- The bold values in Table 4 require further discussion.

- The ‘Discussion’ chapter should be completed.

- The element of novelty - innovation in the 'abstract', the 'introduction' - should be emphasised more.

- The text of the article and the list of literature should be prepared according to the requirements of the journal.

- Literature review is not rigorous. More literature on the safety and GPR.

- How the research subject - 'Combined CNN and RNN neural networks for GPR detection of railway subgrade diseases' - to the accuracy of the GPR depending on vehicle speed?

This completes the review.
Kind regards

Author Response

The publication is distinguished by the reference to own research with noticeable practical experience. The adopted and implemented structure of the publication corresponds well to the goal, which allowed to draw a wide range of conclusions.

Further information on various issues identified in the manuscript appears below:

1 I propose to expand the scientific research literature to include issues in the integration of UAVs with GPR used in rail infrastructure surveys.

There is little literature related to the combined application of UAVs and ground penetrating radar in railroad infrastructure, but their convergent use is a good direction for the future, and we thank you for your suggestion. We cite the literature on the joint application of ground penetrating radar and UAVS in the study of archaeological sites

2 It is proposed to make a tabulation of advantages and disadvantages for the innovative solution.

A tabulation of advantages and disadvantages for the innovative solution has been added.

3 Section 2 - the accuracy of the GPR in terms of monitoring the railway infrastructure should be presented - in the situation where the GPR is built into the train structure. Then we have different speeds. What are the GPR accuracies depending on the speed of the vehicle?

Our data were collected in distance mode with a channel spacing of 11.25 cm. the accuracy of the ground-penetrating radar depends on the size of the channel spacing.

4 The numbering of chapters and subsections needs to be improved throughout the publication. Sometimes identical subsection numbers are used, e.g.: 2.3.

The headings are checked and corrected.

5 The entire text needs improvement in punctuation marks and style.

The punctuation marks and style are corrected

6 The description of Figure 7 should not be in bold.

The problem has been solved.

7 The bold values in Table 4 require further discussion.

Since data labeling is very time-consuming, we only have a limited amount of data, which leads to a lower accuracy of our CRNN network than the Faster RCNN network under a certain IoU threshold. In the subsequent work we continue to collect ground penetrating radar railroad roadbed disease data in addition to optimizing our CRNN network model.

8 The ‘Discussion’ chapter should be completed.

We discussed the CRNN model in Chapter 4, but deep learning network models are difficult and time-consuming to construct, especially in terms of changing network parameters to improve classification accuracy. In the next step we continue to optimize our CRNN model. Thank you for your suggestions!

9 The element of novelty - innovation in the 'abstract', the 'introduction' - should be emphasised more.
The abstract and introduction has been modified.

10 The text of the article and the list of literature should be prepared according to the requirements of the journal.
      The text has been corrected to follow the text style of the journal sample, and the references were rearranged using Zotero.

11 Literature review is not rigorous. More literature on the safety and GPR.
      We have reorganized the references

12 How the research subject - 'Combined CNN and RNN neural networks for GPR detection of railway subgrade diseases' - to the accuracy of the GPR depending on vehicle speed?

Your suggestions are very useful and are the next step we need to consider.

Round 2

Reviewer 4 Report

Dear Authors thanks.